# Structural transformation and the gender pay gap in Sub-Saharan Africa

**Goedele Van den Broeck**[1]*, **Talip Kilic**[2], **Janneke Pieters**[3]

**1** Earth and Life Institute, Université Catholique de Louvain, Louvain-la-Neuve, Belgium, **2** Development Data Group, World Bank, Washington, DC, United States of America, **3** Social Sciences Department, Wageningen University and Research, Wageningen, Netherlands

* Goedele.vandenbroeck@uclouvain.be

## Abstract

The focus of this study is the implications of structural transformation for gender equality, specifically equal pay, in Sub-Saharan Africa. While structural transformation affects key development outcomes, including growth, poverty, and access to decent work, its effect on the gender pay gap is not clear ex-ante. Evidence on the gender pay gap in sub-Saharan Africa is limited, and often excludes rural areas and informal (self-)employment. This paper provides evidence on the extent and drivers of the gender pay gap in non-farm wage- and self-employment activities across three countries at different stages of structural transformation (Malawi, Tanzania and Nigeria). The analysis leverages nationally-representative survey data and decomposition methods, and is conducted separately among individuals residing in rural versus urban areas in each country. The results show that women earn 40 to 46 percent less than men in urban areas, which is substantially less than in high-income countries. The gender pay gap in rural areas ranges from (a statistically insignificant) 12 percent in Tanzania to 77 percent in Nigeria. In all rural areas, a major share of the gender pay gap (81 percent in Malawi, 83 percent in Tanzania and 70 percent in Nigeria) is explained by differences in workers' characteristics, including education, occupation and sector. This suggests that if rural men and women had similar characteristics, most of the gender pay gap would disappear. Country-differences are larger across urban areas, where differences in characteristics account for only 32 percent of the pay gap in Tanzania, 50 percent in Malawi and 81 percent in Nigeria. Our detailed decomposition results suggest that structural transformation does not consistently help bridge the gender pay gap. Gender-sensitive policies are required to ensure equal pay for men and women.

## Introduction

Structural transformation refers to the reallocation of economic activity across sectors of the economy, entailing a shift from employment in agriculture to industry and services [1, 2]. It is closely associated with urbanization, whereby an increasing share of the population lives in towns and cities rather than rural areas. Structural change affects key development outcomes, including growth, poverty, and access to decent work, all of which are at the heart of the Sustainable Development Goals.

**Data Availability Statement:** All relevant data are within the paper and its Supporting Information files.

**Funding:** G.V.d.B. was financially supported by the Research Foundation Flanders (FWO) (FWO post-

doctoral fellowship). The funders had no role in study design, data collection and analysis, decision to publish, or preparation of the manuscript.

**Competing interests:** The authors have declared that no competing interests exist.

The focus of this study is the implications of structural transformation for gender equality, and specifically equal pay. Reducing the gender pay gap is an important prerequisite for achieving gender equality. It is well documented that men earn more than women [3]. This is not only because men are often more educated and experienced, and are more likely to work in higher-paid sectors and occupations, but also due to gender discrimination–that is, men tend to earn more than women even if they perform the same job and have the same education and experience [4].

The effect of structural transformation on the gender pay gap is not clear ex-ante. On the one hand, with increasing education levels and declining gender education gaps, the gender pay gap may shrink if women receive the same returns to education as men [5]. Improved education of women and access to health facilities lead to lower fertility rates, making it easier for women to participate in the labor market [6, 7]. Women's relative productivity might increase with the shift of employment towards services and due to technological change that reduces the physical labor intensity of jobs [8, 9]. Finally, more competition in the labor market due to increased global integration of markets may reduce pay discrimination, as labor markets become more efficient and the opportunity cost of discrimination increases [10].

On the other hand, structural transformation may exacerbate the gender pay gap even if it enhances women's employment opportunities [11]. Occupational and sectoral segregation have increased over time in many countries [12]. Women can end up in lower-paid jobs if social norms retain women in less productive work activities [13] or if their bargaining power vis-à-vis their spouses and employers is limited and they continue to combine work with household responsibilities [14–16].

The implications of structural transformation for the gender pay gap are particularly relevant in Sub-Saharan Africa (SSA). The region has shown early signs of structural transformation since the 2000s [17]. Mounting evidence points to high economic growth, a declining share of people employed in agriculture, and fast urbanization [18]. Despite this progress, the pace of economic development varies substantially across and within countries, as witnessed by persistently high poverty numbers in certain countries, particularly in rural areas [19]. Employment in SSA mainly shifted towards non-tradable services (e.g. personal services and commerce) as opposed to manufacturing or tradable services (e.g. finance, insurance, real estate and business) [20]. This is partly due to the commodity price boom in the 2000s, which spurred the emergence of so-called consumption cities that have been characterized by high shares of (food) imports and employment in non-tradable services [21]. Such structural change has historically been associated with larger income inequality, with potential adverse implications for vulnerable people in general and women in particular [22].

Moreover, SSA is already one of the regions with highest gender inequalities. Women are systematically disadvantaged in access to education, asset ownership and economic opportunities [5, 23–25] Considerable gender pay gaps exist in urban labor markets [26–30] and productivity on female-managed agricultural plots is 20 to 30 percent lower compared to male-managed counterparts [31–35].

Against this background, there is scant systematic evidence on the drivers of the gender pay gap in SSA and its interactions with structural transformation. The available evidence on the gender pay gap in low- and middle-income countries pertains only to urban labor markets or formal wage employment. Without comparable insights in rural areas and in (informal) self-employment, which constitutes the largest source of non-farm employment in SSA [24], it is not possible to fully understand the implications of structural transformation for gender pay gap and to formulate well-targeted policies that address differential extent and sources of gender pay inequality in urban and rural labor markets.

To help fill knowledge gaps, we first provide evidence on the extent and drivers of the gender pay gap in non-farm wage- and self-employment activities in Malawi, Nigeria and Tanzania. We then describe linkages between the gender pay gap and structural transformation through a spatial comparison across and within these three countries. Ideally, structural transformation is investigated by comparing countries over time to capture the development process in the long run. However, this requires extensive longitudinal surveys with high-quality data on individual earnings from wage- and self-employment in both urban and rural areas. Such data are not yet available for Sub-Saharan Africa. Therefore, we opt for a different strategy: we do not only compare countries that are at a different stage in structural transformation, but also compare rural and urban areas in each country. We apply Kitagawa-Oaxaca-Blinder decomposition methods that have been widely used in labor economics to decompose the gender pay gap into (i) a component that is explained by gender differences in observable characteristics, such as education, sector, and occupation, and (ii) another component that is explained by gender differences in returns to these characteristics [36–38]. We pay specific attention to education, sector and occupation, as structural transformation is typically associated with changes in these dimensions. On the whole, we provide important insights into patterns that are applicable to multiple countries, but also reveal country-specific findings.

The choice for including Malawi, Nigeria and Tanzania in our analysis is partly motivated by their different stage in structural transformation and partly based on data availability. First, according to several World Development Indicators [39], Nigeria tends to be at a more advanced stage of structural transformation than Tanzania, followed by Malawi. For example, gross domestic product (GDP) per capita in 2019 was 5,348 int.$ in Nigeria, 2,771 int.$ in Tanzania and 1,104 int.$ in Malawi. Non-agricultural employment accounted for a higher share of total employment in Nigeria (65%), compared to Tanzania (35%) and Malawi (24%). Also urbanization rates in 2019 were highest in Nigeria (51%), followed by Tanzania (35%) and Malawi (17%). All three countries have embarked on the path of structural transformation during the past 20 years. While these indicators were also higher for Nigeria in 2000 with similar initial lower levels for Tanzania and Malawi, growth rates have been strongest in Tanzania and lowest in Malawi.

Second, we leverage household survey data that are representative at the national-level and for urban and rural areas, and that cover extensive, individual-level information on labor earnings and inputs, including wage- and self-employment activities. Such data are in general quite rare for Sub-Saharan Africa, but data collection efforts have steadily improved during the past ten years, both in terms of quantity and quality. An important initiative has been the Living Standards Measurement Study–Integrated Surveys on Agriculture (LSMS-ISA), conducted by the national statistical offices with technical and financial support from the World Bank. While the LSMS-ISA covers multiple countries, the specific data requirements for the purpose of our study are only fulfilled by the data from Malawi, Nigeria and Tanzania.

## Materials and methods

### Data

We use data from the Malawi Fourth Integrated Household Survey 2016/17, the Nigeria General Household Survey–Panel 2015/16, and the Tanzania National Panel Survey 2012/13. The surveys were implemented by the respective national statistical office (NSO) in Malawi, Nigeria and Tanzania. Each survey was implemented under the respective Statistical Act of that country, which governs the roles, responsibilities, and mandates of the NSO in producing and disseminating official statistics that each survey informed. Per the powers granted to the NSOs in fulfilling their duties, the NSO-implemented surveys are not subject to approval by an

Institutional Review Board (IRB) and the NSOs are not required to obtain written informed consent from participants. However, it is established practice to obtain verbal informed consent prior to the face-to-face interview. Refusals can happen, in which case no interview is conducted, but any documentation on this is not available to the authors. The anonymized unit-record survey data and the basic information documents, which provide details regarding questionnaire design, sampling and fieldwork implementation, are publicly available at the World Bank Microdata Library.

We use these specific waves as they were the most recent available at the time of the analysis. In case of Tanzania, the wave of 2014/15 could not be used because data on annual hours were missing for 39% of workers (the self-employed). In the meantime, more recent waves have become available (Malawi Fifth Integrated Household Survey 2019/20, Nigeria General Household Survey–Panel 2018/19 and Tanzania National Panel Survey 2019/20). While it is possible to add these waves as additional data points in our analysis, the time between surveys is still limited (at most 7 years in the case of Tanzania). Over this time span, labor force participation rates were virtually constant and the agricultural share of employment declined only slightly, according to the ILO modelled estimates. Hence, we would not be getting much additional insight into the link between structural change and the gender wage gap. Finally, we initially considered other countries that are included in the LSMS-ISA initiative as well, such as Ethiopia, Uganda, Niger, Mali and Burkina Faso. However, we found that the quality of data on earnings and inputs in non-farm employment was not sufficient for these countries, due to missing individual-level information for self-employment or no distinction between wage and self-employment.

The surveys are designed to be representative at the national, rural and urban level, and are based on multistage stratified sampling strategies. Primary sampling units are enumeration areas (EAs), which are small geographical areas defined by national censuses. In the first sampling stage, around 400 EAs in Nigeria and Tanzania and 800 EAs in Malawi are randomly selected. In the second stage, around 10 to 30 households are randomly selected per EA. Stratification is based on regional zones separated into urban and rural areas. Areas are categorized to be urban or rural as determined by the national statistical offices. This definition reflects the specific context and depends on countries' official administrative criteria. In Tanzania, regional and district headquarters are always considered as urban, while the urban status of any EA beyond these administrative boundaries is determined subjectively by local census committees. In Nigeria, all state capitals and settlements with populations of at least 20,000 are categorized as urban. In Malawi, urban centers are defined according to levels of service provision, such as administration, commerce and business, health, education and infrastructure, and are classified into two primary centers (Blantyre and Lilongwe), one regional center (Mzuzu) and six sub-regional centers (Karonga, Liwonde, Mangochi, Salima, Dedza and Bangula).

The surveys supported by the LSMS-ISA initiative rely on multi-topic household questionnaires that are comparable across countries with some questions adapted to the local context. The questionnaires include modules on demographics, education, health, and labor, which elicit information at the individual-level. The module on labor is administered to all household members over age five and is complemented by a module on non-farm enterprises, which, for each household enterprise that was operated in the last 12 months prior to the interview date, identifies enterprise owners and collects information about enterprise activities, household and hired labor inputs, revenues and costs. We limit the sample to men and women aged 25 to 55 who are currently not enrolled in school. We retain 23,827 observations in rural areas and 9,179 observations in urban areas.

## Measurement of hourly pay

We define non-farm employment as wage and self-employment that takes place outside the agricultural, fishery and forestry sectors. If a person is involved in multiple jobs, we analyze the job with the highest labor input. Employment for safety net / public works programs and *ganyu* labor (in case of Malawi) are not considered. Household members who work for a household enterprise but are not the owner are not included. Participation in non-farm employment is based on the 12 months prior to the survey to avoid seasonality effects that are common in these countries. We focus specifically on non-farm employment, as opposed to overall employment, for the main reason that individual earnings tied to household farms are very hard (if not impossible) to estimate. For instance, the survey questions to elicit data on revenues and costs are typically asked at different levels: while the questions on costs of pesticides, fertilizer and labor are asked at the plot-level, the information regarding costs of seeds and revenues from market sales are elicited at the crop-level, and the data on rental/depreciating costs of farm equipment are collected at the farm-level. Arguably more challenging is the allocation of net household agricultural income across household laborers, as it would be highly sensitive to the choice of the survey questions that are used for this purpose (e.g., questions that capture owners versus managers versus laborers at the plot- and livestock group-level). In addition, a lot of the work done in farm employment is unpaid, with the resulting production destined for own consumption, which is especially the case for women's work [40] and further complicates the estimation of individual earnings and gender pay gaps.

Hourly pay is calculated as annual earnings divided by annual hours. Annual earnings are converted to real international dollars using purchasing power parities (PPP) and consumer price indices. Annual earnings of wage employment are net income from cash and in-kind payments. Annual earnings of self-employment are reported net profits (gross revenues minus costs for hired labor and other business costs) per enterprise. If there are multiple owners, the earnings are distributed across the owners proportional to their labor input. This assumption applies to few enterprises (13% in Malawi and 4% in Nigeria and Tanzania) and does not affect our main results. We take the natural logarithm of hourly pay and winsorize its distribution at the top 1% to control for outliers.

We truncate unrealistically high values of labor input to a maximum of 4,368 hours per year, assuming that no one can work more than 12 hours per day during 7 days per week. If hours per day are not reported and payment period is expressed per day, we assume that a working week consists of six days. We replace missing values for labor earnings and hours by median values. Data quality is in general high and few observations need to be cleaned (S1 Table).

## Decomposition analysis

We decompose the gender pay gap in three steps. First, we analyze Mincer earnings functions of the following type:

$$Y_{G,i} = \beta_0 + \sum_{k=1}^{K} X'_{G,ik}\beta_{G,k} + \varepsilon_{G,i} \tag{1}$$

where $Y_{Gi}$ is the logarithm of hourly pay of individual $i$ in sample $G$ that is pooled (*), female (*F*) or male (*M*). $X$ is a vector of $k$ characteristics that influence individual $i$'s hourly pay. The pooled sample includes a gender dummy.

Pay determinants cover human capital, sector and occupation. Human capital includes education defined as primary, secondary or tertiary degree with no degree as base level, potential experience and its square defined as age minus six years minus years of education, and a dummy if involved in multiple jobs. Sector is classified according to the International Standard

Industry Codes (ISIC) and includes dummies for mining, manufacturing, electricity / utilities, construction, transportation / storage / communication, finance/real estate and other services (accommodation & food service, ICT administrative, human health and social work) with commerce as base level. Occupation is classified according to the International Standard Classification of Occupation (ISCO) and includes dummies for self-employment with household workers (irrespective of the duration and payment of these workers), low-skilled wage employment (corresponding to elementary occupations), medium-skilled wage employment (corresponding to clerical support workers, service and sales workers, skilled agricultural, forestry and fishery workers, craft and related trades workers, plant and machine operators and assemblers, and armed forces) and high-skilled wage employment (corresponding to managers, professionals, technicians and associate professionals) with self-employment without household workers as base level. Missing values for sector and education are coded as separate category.

We control for the survey context by including dummies for region, month of interview, enumerator and whether individuals responded themselves. We apply survey weights and control for the stratified sampling strategy to derive representative results. We analyze these models for the pooled countries and each country separately across rural and urban areas.

Second, we define the gender pay gap $D$ as follows:

$$D = E[Y_M] - E[Y_F] \qquad (2)$$

where *E[Y]* denotes the expected value of the logarithm of hourly pay of men (*M*) or women (*F*). The gender pay gap is calculated in log points and is exponentiated to express it in percent.

Third, we follow the Kitagawa-Oaxaca-Blinder decomposition method and decompose the gender pay gap into a part that is explained by differences in characteristics (i.e. endowment effect) and a part that is explained by differences in returns to these characteristics (i.e. structural effect) [36–38]:

$$D = \sum_{k=1}^{K} (E[X_{M,k}] - E[X_{F,k}])\hat{\beta}_{*,k} + \left(\sum_{k=1}^{K} E[X_{M,k}](\hat{\beta}_{M,k} - \hat{\beta}_{*,k}) + \sum_{k=1}^{K} E[X_{F,k}](\hat{\beta}_{*,k} - \hat{\beta}_{F,k})\right) (3)$$

where *E[X]* denotes the expected value and $\hat{\beta}_{G,k}$ the estimated coefficient on the *k* characteristics in the earnings functions for sample *G*. We use the pooled coefficients to impose no assumptions on positive or negative discrimination towards men or women [41].

## Results

### Characteristics of non-farm employment

In Malawi, Nigeria, and Tanzania, men are more likely to be non-farm employed than women (Table 1). The gender gap in non-farm employment is consistent across and within countries, and whether the survey recall period is seven days or 12 months. Non-farm employment rates vary substantially across countries, with rates highest in Nigeria and lowest in Malawi, which is in line with their different stage in structural transformation. Non-farm employment rates are higher in urban areas than in rural areas, both for men and women, but the gender gap in non-farm employment is larger in urban areas.

Men and women in non-farm employment differ in terms of human capital and the types of jobs they perform (Table 2 for rural workers and Table 3 for urban workers). Women are less educated than men, both in rural and urban areas. While education levels are much lower in rural areas, gender disparities are larger. The gender education gap is relatively low in rural and urban Tanzania. Our earnings functions (S2 Table for rural workers and S3 Table for urban workers) show that earnings from secondary and especially tertiary education are

**Table 1. Non-farm employment rates for population aged 25–55 in Malawi, Tanzania and Nigeria.**

| | Malawi | | | Tanzania | | | Nigeria | | |
|---|---|---|---|---|---|---|---|---|---|
| | *Women* | *Men* | *P* | *Women* | *Men* | *P* | *Women* | *Men* | *P* |
| *Rural* | | | | | | | | | |
| NF empl. rate (7d) | 0.14 | 0.26 | < .001 | 0.25 | 0.40 | < .001 | 0.42 | 0.47 | .005 |
| NF empl. rate (12m) | 0.18 | 0.30 | < .001 | 0.27 | 0.42 | < .001 | 0.42 | 0.46 | .053 |
| Observations | 7,509 | 6,530 | | 2,341 | 2,087 | | 3,147 | 2,210 | |
| *Urban* | | | | | | | | | |
| NF empl. rate (7d) | 0.39 | 0.64 | < .001 | 0.49 | 0.78 | < .001 | 0.64 | 0.77 | < .001 |
| NF empl. rate (12m) | 0.49 | 0.67 | < .001 | 0.58 | 0.82 | < .001 | 0.62 | 0.75 | < .001 |
| Observations | 2,138 | 2,056 | | 1,394 | 1,292 | | 1,327 | 972 | |

Notes: NF empl. rate means non-farm employment rate, with recall period between parentheses. Population statistics are corrected using sampling weights. P-values are derived from bivariate regressions.

**Table 2. Descriptive statistics across gender for non-farm employed people aged 25–55 in rural Malawi, Tanzania and Nigeria.**

| | Malawi | | | Tanzania | | | Nigeria | | |
|---|---|---|---|---|---|---|---|---|---|
| | *Women* | *Men* | *P* | *Women* | *Men* | *P* | *Women* | *Men* | *P* |
| *Human capital* | | | | | | | | | |
| No degree | 0.67 | 0.53 | < .001 | 0.30 | 0.23 | 0.012 | 0.55 | 0.34 | < .001 |
| Primary degree | 0.14 | 0.12 | 0.316 | 0.64 | 0.67 | 0.194 | 0.24 | 0.23 | 0.813 |
| Secondary degree | 0.16 | 0.29 | < .001 | 0.06 | 0.09 | 0.048 | 0.18 | 0.36 | < .001 |
| Tertiary degree | 0.03 | 0.06 | 0.014 | 0.01 | 0.01 | 0.783 | 0.03 | 0.08 | < .001 |
| Age | 36.31 | 37.02 | 0.143 | 37.65 | 37.66 | 0.983 | 38.28 | 41.29 | < .001 |
| Potential experience | 23.87 | 23.16 | 0.187 | 25.90 | 25.28 | 0.265 | 27.91 | 28.26 | 0.544 |
| Multiple jobs | 0.02 | 0.03 | < .001 | 0.09 | 0.11 | < .001 | 0.27 | 0.22 | < .001 |
| *Occupation* | | | | | | | | | |
| Self-empl. without family labor | 0.62 | 0.48 | < .001 | 0.75 | 0.50 | < .001 | 0.71 | 0.60 | < .001 |
| Self-empl. with family labor | 0.25 | 0.12 | < .001 | 0.08 | 0.07 | 0.550 | 0.19 | 0.12 | < .001 |
| Low-skilled employee | 0.03 | 0.05 | 0.054 | 0.05 | 0.13 | < .001 | 0.00 | 0.01 | 0.047 |
| Medium-skilled employee | 0.02 | 0.20 | < .001 | 0.06 | 0.23 | < .001 | 0.03 | 0.10 | < .001 |
| High-skilled employee | 0.07 | 0.13 | < .001 | 0.05 | 0.06 | 0.263 | 0.07 | 0.16 | < .001 |
| Missing occupation | 0.01 | 0.02 | 0.419 | 0.01 | 0.01 | 0.262 | 0.00 | 0.01 | 0.012 |
| *Sector* | | | | | | | | | |
| Mining | 0.00 | 0.01 | 0.190 | 0.02 | 0.03 | 0.103 | 0.00 | 0.01 | 0.408 |
| Manufacturing | 0.22 | 0.13 | < .001 | 0.12 | 0.10 | 0.424 | 0.20 | 0.14 | < .001 |
| Electricity, utilities | 0.00 | 0.01 | 0.060 | 0.00 | 0.00 | 0.099 | 0.00 | 0.00 | 0.736 |
| Construction | 0.01 | 0.02 | 0.055 | 0.01 | 0.16 | < .001 | 0.02 | 0.08 | < .001 |
| Commerce | 0.60 | 0.43 | < .001 | 0.57 | 0.42 | < .001 | 0.52 | 0.35 | < .001 |
| Transport, storage, comm. | 0.01 | 0.07 | < .001 | 0.00 | 0.06 | < .001 | 0.04 | 0.11 | < .001 |
| Finance, real estate | 0.00 | 0.00 | 0.191 | 0.00 | 0.01 | 0.075 | 0.00 | 0.01 | 0.124 |
| Other services | 0.14 | 0.26 | < .001 | 0.25 | 0.18 | 0.011 | 0.22 | 0.30 | < .001 |
| Missing sector | 0.02 | 0.08 | 0.002 | 0.04 | 0.04 | 0.897 | 0.00 | 0.01 | 0.060 |
| Observations | 1,279 | 1,877 | | 657 | 935 | | 1,293 | 992 | |

Notes: Population statistics are corrected using sampling weights. P-values are derived from bivariate regressions.

**Table 3. Descriptive statistics across gender for non-farm employed people aged 25–55 in urban Malawi, Tanzania and Nigeria.**

| | Malawi | | | Tanzania | | | Nigeria | | |
|---|---|---|---|---|---|---|---|---|---|
| | *Women* | *Men* | *P* | *Women* | *Men* | *P* | *Women* | *Men* | *P* |
| *Human capital* | | | | | | | | | |
| No degree | 0.41 | 0.25 | < .001 | 0.15 | 0.13 | 0.340 | 0.17 | 0.07 | < .001 |
| Primary degree | 0.10 | 0.10 | 0.991 | 0.63 | 0.60 | 0.337 | 0.28 | 0.26 | 0.325 |
| Secondary degree | 0.33 | 0.48 | < .001 | 0.20 | 0.24 | 0.124 | 0.46 | 0.50 | 0.323 |
| Tertiary degree | 0.15 | 0.17 | 0.559 | 0.02 | 0.03 | 0.091 | 0.09 | 0.18 | < .001 |
| Age | 35.44 | 36.81 | 0.007 | 36.16 | 36.57 | 0.341 | 39.63 | 40.60 | 0.082 |
| Potential experience | 20.44 | 20.41 | 0.949 | 22.75 | 22.71 | 0.932 | 24.82 | 24.39 | 0.492 |
| Multiple jobs | 0.08 | 0.09 | 0.575 | 0.19 | 0.21 | 0.282 | 0.32 | 0.33 | 0.890 |
| *Occupation* | | | | | | | | | |
| Self-empl. without family labor | 0.46 | 0.26 | < .001 | 0.64 | 0.42 | < .001 | 0.63 | 0.56 | 0.012 |
| Self-empl. with family labor | 0.15 | 0.06 | < .001 | 0.03 | 0.03 | 0.908 | 0.13 | 0.06 | < .001 |
| Low-skilled employee | 0.11 | 0.07 | 0.013 | 0.06 | 0.12 | < .001 | 0.01 | 0.01 | 0.331 |
| Medium-skilled employee | 0.14 | 0.43 | < .001 | 0.19 | 0.32 | < .001 | 0.07 | 0.15 | 0.001 |
| High-skilled employee | 0.13 | 0.15 | 0.189 | 0.07 | 0.12 | 0.009 | 0.15 | 0.21 | 0.022 |
| Missing occupation | 0.01 | 0.01 | 0.418 | 0.00 | 0.00 | 0.575 | 0.01 | 0.02 | 0.949 |
| *Sector* | | | | | | | | | |
| Mining | 0.00 | 0.00 | 0.925 | 0.02 | 0.04 | 0.017 | 0.00 | 0.01 | 0.041 |
| Manufacturing | 0.11 | 0.10 | 0.641 | 0.08 | 0.10 | 0.242 | 0.13 | 0.13 | 0.801 |
| Electricity, utilities | 0.01 | 0.01 | 0.221 | 0.01 | 0.01 | 0.653 | 0.01 | 0.01 | 0.812 |
| Construction | 0.00 | 0.03 | 0.001 | 0.01 | 0.09 | < .001 | 0.02 | 0.07 | 0.001 |
| Commerce | 0.47 | 0.26 | < .001 | 0.42 | 0.32 | 0.001 | 0.46 | 0.26 | < .001 |
| Transport, storage, comm. | 0.03 | 0.14 | < .001 | 0.01 | 0.14 | < .001 | 0.05 | 0.11 | < .001 |
| Finance, real estate | 0.01 | 0.01 | 0.923 | 0.01 | 0.02 | 0.224 | 0.01 | 0.02 | 0.847 |
| Other services | 0.30 | 0.34 | 0.198 | 0.42 | 0.25 | < .001 | 0.31 | 0.39 | 0.001 |
| Missing sector | 0.07 | 0.10 | 0.135 | 0.02 | 0.04 | 0.252 | 0.00 | 0.00 | 0.259 |
| Observations | 1,131 | 1,462 | | 771 | 1,043 | | 804 | 711 | |

Notes: Population statistics are corrected using sampling weights. P-values are derived from bivariate regressions.

significantly higher than having no or only primary education. This is a first indication that gender education disparities may increase with structural transformation, thereby exacerbating the gender pay gap.

In terms of occupation, the majority of nonfarm workers are self-employed. Women are substantially more likely to be self-employed than men, especially in rural areas, while men are more likely to be in medium-skilled and high-skilled wage employment. Our earnings functions show that earnings from rural self-employment are significantly lower than any type of rural wage employment. Since structural change from self-employment to wage employment within rural areas seems to offer opportunities almost exclusively to men, this is likely to exacerbate gender pay inequality. In urban areas, only high-skilled wage employment generates higher earnings than self-employment, so both gender differences in occupation and pay differences across occupations are less likely to contribute to gender pay inequality.

Regarding the sectors of work, rural and urban nonfarm employment is highly concentrated in services, while manufacturing and industry account for only 10 to 20% of employment. This distribution illustrates the growth without industrialization that characterizes much of SSA's recent development. In rural areas, women are more likely to work in manufacturing than men. Within services, commerce (retail and wholesale trade) is by far the

most important sector of work, especially for women. The second most important sector is other services, which includes the public sector and is more important in urban areas compared to rural, and for men compared to women (except in Tanzania). Our earnings functions show that hourly earnings in commerce and in other services do not differ significantly from those in manufacturing in any of the countries. Earnings are significantly higher only in mining and in construction (in both rural and urban areas), and in transport, storage and communication and in finance and real estate in urban areas. These sectors are relatively small, but they are dominated by men, especially in Tanzania.

## Gender pay gap in non-farm employment

We find that non-farm employed women earn substantially less than men in all three countries (Fig 1, with more detailed regression results in Table 4). The gender gap in urban areas varies from 40 percent in Nigeria to 46 percent in Malawi and Tanzania. The variation across rural areas is much wider, where it ranges from (a statistically insignificant) 12 percent in Tanzania over 44 percent in Malawi to 77 percent in Nigeria. In all rural areas, a major share of the gender pay gap (81 percent in Malawi, 83 percent in Tanzania and 70 percent in Nigeria) is explained by differences in workers' characteristics, including education, occupation and sector. This suggests that if rural men and women had similar characteristics, most of the gender pay gap would disappear. This is also reflected in the small and sometimes insignificant

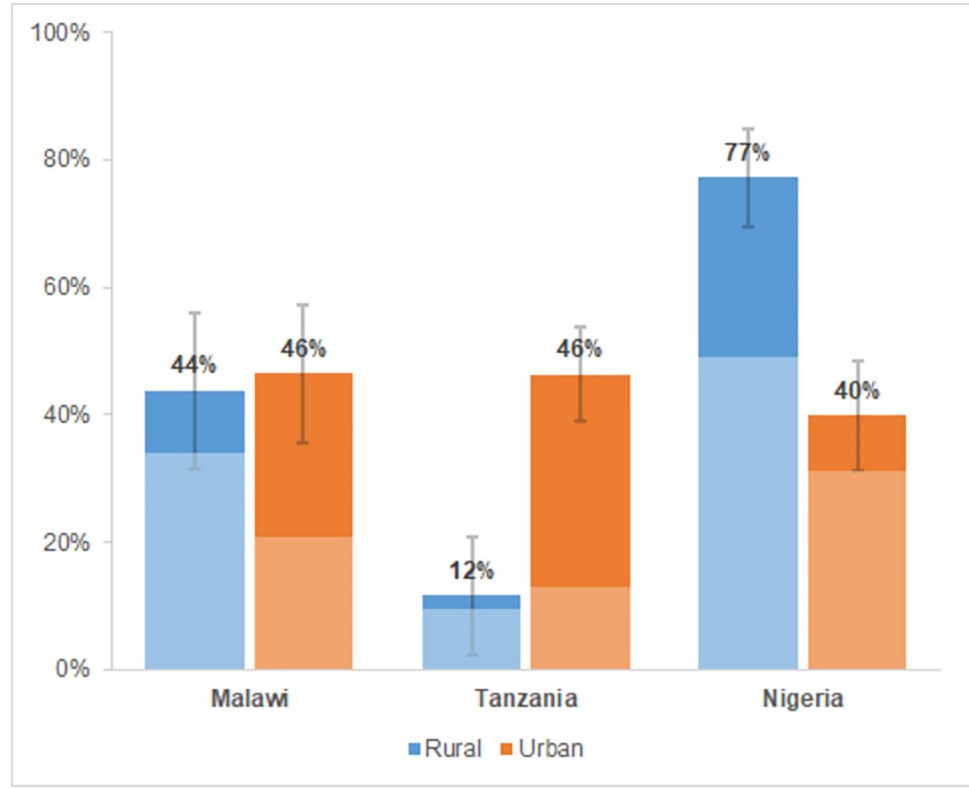

**Fig 1. Average gender pay gap (in %) in non-farm employment across rural and urban areas in Malawi, Tanzania and Nigeria for women and men aged 25–55.** The lighter (darker) area represents the part that is explained by differences in characteristics (returns). Error bars represent standard errors of the mean. Results are obtained from regressions presented in Table 4. Population statistics are corrected using sampling weights. The number of observations for the different bars are as follows: 3,156; 2,593; 1,592; 1,814; 2,285; 1,515.

**Table 4. Kitagawa-Oaxaca-Blinder decomposition of gender pay gap for non-farm employed people aged 25–55 in Malawi, Tanzania and Nigeria.**

| | Malawi | | Tanzania | | Nigeria | |
|---|---|---|---|---|---|---|
| | *Rural* | *Urban* | *Rural* | *Urban* | *Rural* | *Urban* |
| *Mean gender pay gap* | 0.362*** | 0.381*** | 0.110 | 0.380*** | 0.572*** | 0.336*** |
| | (0.115) | (0.102) | (0.088) | (0.071) | (0.073) | (0.084) |
| *A. Aggregate decomposition* | | | | | | |
| Endowment effect | 0.293*** | 0.188*** | 0.091** | 0.123*** | 0.399*** | 0.272*** |
| | (0.060) | (0.059) | (0.047) | (0.037) | (0.054) | (0.059) |
| Endowment effect (share) | 80.85% | 49.43% | 82.93% | 32.43% | 69.72% | 81.09% |
| Structural effect | 0.069 | 0.193** | 0.019 | 0.257*** | 0.173** | 0.063 |
| | (0.122) | (0.088) | (0.088) | (0.067) | (0.071) | (0.071) |
| Structural effect (share) | 19.15% | 50.57% | 17.07% | 67.57% | 30.28% | 18.91% |
| *B. Detailed decomposition* | | | | | | |
| *B1. Endowment effect* | | | | | | |
| Education (aggregated) | 0.116*** | 0.045 | 0.033** | 0.037** | 0.109*** | 0.123*** |
| | (0.037) | (0.033) | (0.014) | (0.016) | (0.034) | (0.033) |
| Experience (aggregated) | -0.008 | 0.000 | -0.009 | 0.011 | 0.041** | 0.029* |
| | (0.008) | (0.006) | (0.009) | (0.009) | (0.017) | (0.017) |
| Sector (aggregated) | -0.048 | 0.072** | 0.096*** | 0.095*** | 0.027 | 0.070** |
| | (0.046) | (0.032) | (0.032) | (0.023) | (0.032) | (0.030) |
| Occupation (aggregated) | 0.245*** | 0.093** | -0.031 | -0.023 | 0.198*** | 0.064* |
| | (0.059) | (0.044) | (0.035) | (0.023) | (0.032) | (0.034) |
| Context (aggregated) | -0.013 | -0.021 | 0.002 | 0.002 | 0.024 | -0.014 |
| | (0.036) | (0.023) | (0.023) | (0.017) | (0.023) | (0.023) |
| *B2. Structural effect* | | | | | | |
| Education (aggregated) | 0.142 | 0.055 | -0.332* | -0.021 | 0.134** | -0.094* |
| | (0.173) | (0.041) | (0.171) | (0.092) | (0.067) | (0.057) |
| Experience (aggregated) | -0.300 | 0.209 | -0.790 | -0.057 | -1.052** | -0.344 |
| | (0.649) | (0.364) | (0.541) | (0.370) | (0.461) | (0.407) |
| Sector (aggregated) | -0.238 | -0.317** | 0.346*** | 0.085 | 0.132 | -0.158* |
| | (0.307) | (0.161) | (0.118) | (0.094) | (0.144) | (0.094) |
| Occupation (aggregated) | 0.353 | -0.167* | 0.145 | -0.106 | -0.550*** | -0.121 |
| | (0.261) | (0.100) | (0.144) | (0.168) | (0.115) | (0.110) |
| Context (aggregated) | 1.038** | -0.261 | -0.601*** | -0.125 | -0.065 | -0.031 |
| | (0.471) | (0.202) | (0.224) | (0.098) | (0.122) | (0.120) |
| Constant | -0.925 | 0.673 | 1.251** | 0.480 | 1.575*** | 0.812* |
| | (0.849) | (0.443) | (0.584) | (0.418) | (0.503) | (0.425) |
| Observations | 3,156 | 2,593 | 1,592 | 1,814 | 2,285 | 1,515 |

Notes: Population statistics are corrected using sampling weights. Significant coefficients are indicated with

* p<0.1

** p<0.05 and

*** p<0.01 and expressed in log-points. Standard errors are reported between parentheses.

coefficient for gender in the earnings functions that control for these characteristics (S2 Table). The variation in the endowment effect is much wider across urban areas, where differences in characteristics account for only 32 percent of the pay gap in Tanzania, 50 percent in Malawi and 81 percent in Nigeria.

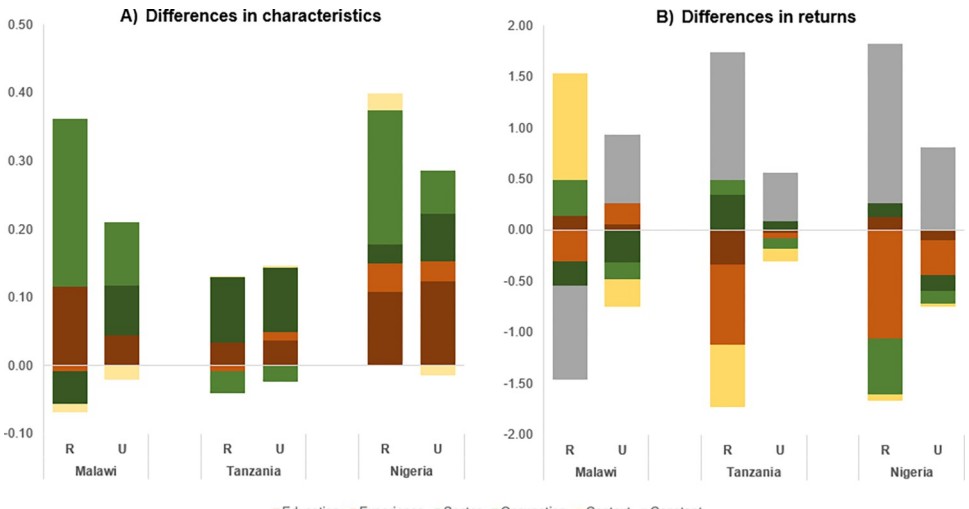

**Fig 2.** Detailed decomposition of average gender pay gap (in log points) in non-farm employment, identifying the relative contributions of gender differences in observable characteristics (Panel A) versus the relative contributions of gender differences in returns to these characteristics (Panel B), across rural (R) and urban (U) areas in Malawi, Tanzania and Nigeria for women and men aged 25–55. Education comprises no, primary, secondary and tertiary schooling; experience comprises experience, its square and multiple jobs; sector comprises mining, manufacturing, electricity/utilities, construction, commerce, transport/storage/communication, finance/real estate, other services and missing sector; occupation comprises self-employed with/without family labor, low-skilled/medium-skilled/high-skilled wage employed; context comprises region, proxy respondent, enumerator and month of interview; and constant comprises all unexplained differences. Results are obtained from regressions presented in Table 4. Population statistics are corrected using sampling weights. The number of observations for the different bars are as follows: 3,156; 2,593; 1,592; 1,814; 2,285; 1,515.

Fig 2 decomposes the gender pay gap further (with more detailed regression results in Table 4). Panel A presents the contributions of gender differences in observable characteristics to the gender pay gap, while Panel B demonstrates the contributions of gender differences in returns to these characteristics. The contributions can take on both negative and positive values, decreasing or exacerbating the gender pay gap, respectively.

Panel A reveals that the contributions of gender differences in education, occupation and sector towards the gender pay gap vary both within and across countries. A common finding is that inequality in education contributes to higher gender pay gaps everywhere, but gender differences in sector and occupation contribute as least as much, and in most cases substantially more. In rural Malawi and Nigeria, gender differences in occupation constitute the most important factor widening the gender pay gap, while in urban areas occupation and sector are equally important. In Tanzania, occupational sorting slightly narrows the gender gap, while sectoral sorting is the main factor. We do not find evidence for gender differences in experience contributing to the gender pay gap.

Panel B shows the part of the gender pay gap that is explained by gender differences in returns to these characteristics. This might stem from pay discrimination but also from unobserved motivation and skills that may differ between men and women. This is reflected in the relatively large contribution by the constant, which captures differences in unobserved determinants of men's and women's pay. The pattern is less clear than for the differences in characteristics: sizes of coefficients are larger but so are standard errors, resulting in the overwhelming majority of these estimates not being statistically significant.

## Robustness checks

We analyze two additional models to check the robustness of the decomposition analysis. First, we use Heckman selection models to control for non-random selection into non-farm employment. In the first stage, we use probit models to estimate the probability of being non-farm employed, for the entire sample of individuals age 25–55 (including those not employed and those working in farming). We control for education, experience and survey settings. We include marital status, number of children, landholdings and livestock units as selection variables. Theory and other studies suggest that these are major determinants of non-farm employment status but are less likely to influence hourly pay directly [42]. Adjusted Wald tests reject the null hypothesis that these selection variables can jointly be excluded in all the regression models, indicating that they are good predictors of participation in non-farm employment (S4 and S5 Tables). We obtain qualitatively similar results when only marital status and landholdings, the strongest predictors for non-farm employment, are used as selection variables. In the second stage, we decompose the gender pay gap using the adjusted earnings functions. We use maximum likelihood estimations and take into account the household survey design. We find that selection bias, as indicated by ρ in S6 and S7 Tables, is either insignificant in rural and urban Malawi and Tanzania or moves in the same direction for both men and women in rural Nigeria, and hence does not influence the estimated gender pay gap. We do find a negative selection bias for rural men in Nigeria, leading to a smaller adjusted pay gap of 43 percent compared to 77 percent in our main results. We therefore conclude that most of our results are robust to non-random participation in non-farm employment.

Our selection variables may not be perfectly exogenous to hourly pay estimates. Thus, as second robustness check, we analyze recentered influence functions to decompose the gender pay gap at the median [43]. Median values are less susceptible to skewed distributions. Moreover, if selection into non-farm employment is non-random, recentered influence functions only require assumptions on the position of the missing pay observations with respect to the median, rather than assumptions on the exogeneity of selection variables [44]. We do not find statistically significant differences between the average and median pay gap and their decompositions (S9 Table), again suggesting that our estimates are robust.

## Discussion

We find substantial gender pay gaps in non-farm employment in rural and urban areas in Malawi, Nigeria and Tanzania. The gap varies slightly across these three countries in urban areas (ranging from 46 percent in Malawi and Tanzania to 40 percent in Nigeria), but the results pertaining rural pay gap show more variation across countries. There is a small (statistically insignificant) pay gap in rural Tanzania, while the gap stands at 44 percent for rural Malawi and 77 percent for rural Nigeria. Thus, the gender pay gap is not always lower in urban areas or in countries that are at a more advanced stage of structural change. This implies that structural transformation does not consistently help bridge the gender pay gap. Below, we explain our results and highlight implications for policy making and future research.

When comparing urban areas across countries, the gender pay gap is lower in Nigeria compared to the other two countries. Nigeria is characterized by a more economically developed labor market. At the same time, our estimated pay gaps are higher than those in high-income countries, where they typically range between 3 and 35 percent [45]. Does this imply that gender pay gaps will be automatically reduced in urban economies, as structural change continues?

This is not necessarily the case. First, as is clear from Fig 2, the gender gap due to differences in characteristics is largest in Nigeria and this is mainly because of differences in educational

attainment. While education levels are higher in Nigeria and there is no gender inequality in primary or secondary degrees, men are twice as likely to obtain a tertiary degree in urban Nigeria. Tertiary education is associated with significantly higher wages, so men's advantage in terms of tertiary education increases the gender pay gap. Unless the returns to tertiary education decline with structural change, which seems unlikely given that structural change typically leads to significant increases in the skill premium [46], further development in urban labor markets may thus increase the gender pay gap, unless the gender gap in tertiary education can be reduced.

Second, zooming in on the types of jobs that men and women perform, the pattern of structural transformation in urban Nigeria is quite different compared to the other countries. Considering occupations, there has not been a clear shift from self-employment to wage employment–in fact, self-employment is more common in urban Nigeria than in the other two countries. At the same time, wage employment in urban Nigeria is concentrated in high-skilled employment, which is dominated by men. In contrast to low- and medium-skilled employment, the returns to high-skilled wage employment are consistently higher than returns to self-employment. This suggests that structural change accompanying further economic growth in Nigeria may manifest itself mainly in the growth of high-skilled employment, where men tend to be overrepresented (which is related to men's advantage in tertiary education), thus leading to higher gender pay gaps. Considering sectors, there are some differences between the countries in terms of the sectoral distribution of employment, but no clear sign of employment shifting to higher productivity service sectors (e.g. finance/real estate, other services) in Nigeria. This is in line with other studies that have observed a shift towards non-tradable services in SSA, as opposed to other developing regions [20]. The lack of a shift in employment in Nigeria might be particularly explained by the country's high reliance on oil exports. In addition, sectoral segregation by gender and its contribution to the gender pay gap is quite comparable across the three countries.

Turning to the rural areas of these countries, the patterns in terms of cross-country differences as well as within-country rural-urban differences are less clear. The gender pay gap is highest in rural Nigeria and is nearly twice the estimate for urban Nigeria. This is probably due to the higher likelihood of men with high-pay characteristics participating in non-farm employment in rural Nigeria. Adjusting for this non-random participation lowers the pay gap to 43 percent, which is more consistent with the patterns observed in Malawi. On the other hand, in rural Tanzania, the pay gap is small and statistically insignificant. Here, educational attainment is almost equal between men and women and is in strong contrast with Malawi and Nigeria where education disparities are much higher and account for about 10 percent of the gender pay gap. Attaining secondary and tertiary education is still rare in rural Tanzania. It is possible that with further educational expansion, gender inequality in educational attainment will increase, leading to a higher gender pay gap.

In addition, while occupational segregation in rural areas is similar across countries, with women overrepresented in self-employment, this does not increase the gender gap in Tanzania because men are mainly concentrated in low- and medium-skilled employment, where wages are not different from those in self-employment. This is different in Malawi and Nigeria, where wage employment at all skill levels commands higher earnings than self-employment and therefore occupational segregation contributes a lot to the gender pay gap in rural areas. So structural transformation within the rural non-farm sector could lead to higher rural gender pay gaps, unless women's access to wage employment in growing sectors improves. Sectoral segregation is the main factor explaining the pay gap in rural Tanzania, with men overrepresented in construction, which is the highest paying sector.

Structural change within rural areas also entails a shift from farm to non-farm activities. In view of the existing estimates of the productivity differences on female-managed and male-

managed agricultural plots of 25 percent in Malawi [33], 35 percent in North Nigeria [34] and 29 percent in Tanzania [35], we conclude that this shift implies a reduction of the gender pay gap in Tanzania, but an increase in Malawi and (especially) Nigeria. While the rural non-farm economy has often been promoted to diversify income sources and reduce poverty, our results point to the need to consider gender-sensitive policies in this shift to ensure that women benefit equally from improvements in overall welfare.

Another important observation emerges when we compare our results with previous estimates of gender pay gaps in low- and middle-income countries [47]. These are typically much lower, which is most likely due to differences in the survey data used for estimation. Such studies often estimate the pay gap based on formal, paid work, probably because of data limitations. Yet, the majority of workers in poorer countries are active in informal, self-employed jobs. Especially women are more likely to work in these lower-paid activities. Failure to include these jobs in the analysis may lead to an underestimation of the gender pay gap, obscuring the urgent need to promote equal pay for men and women. By using the LSMS-ISA data, which include information on all types of employment, we demonstrate the importance of taking informal work into account as well.

We need to acknowledge that a large part of the gender pay gap remains unexplained, both in rural and urban areas. This might be due to the omission of cognitive and non-cognitive skills in our analysis. While the LSMS-ISA data represent a big improvement for data availability and quality in SSA, the surveys do not include modules on skills or training. Returns to different types of skills can be high, however, and gender differences in skills (and their returns) could be an important driver of gender pay gaps [3].

In sum, our results suggest that the gender pay gap is unlikely to decrease mechanically when these countries continue their path of structural transformation. Additional policies are required to ensure equal pay for men and women in non-farm jobs. Priority should be given to reduce inequalities in (higher) educational attainment, but education alone will not be sufficient to cause a gender-inclusive shift in the sectoral and occupational structure of employment, which constitutes the major part of the pay gap. Other factors, including self-efficacy and male support networks, may help reduce occupational and sectoral segregation [48–50]. How the gender pay gap will continue to evolve also depends on how labor markets will change during the COVID-19 pandemic, which is entrenching preexisting labor market gender inequality [51]. Women's jobs have been more affected than men's, because they are more likely to be informally (self-)employed and more involved in childcare, but it is unclear to what extent the crisis will lead to even bigger pay gaps.

## Supporting information

**S1 Table. Data cleaning of annual hours and earnings of non-farm employment in Malawi, Tanzania and Nigeria.**
(DOCX)

**S2 Table. Earnings functions of non-farm employment for men, women and pooled sample in rural Malawi, Tanzania and Nigeria with log hourly pay (real int. $) as dependent variable.**
(DOCX)

**S3 Table. Earnings functions of non-farm employment for men, women and pooled sample in urban Malawi, Tanzania and Nigeria with log hourly pay (real int. $) as dependent variable.**
(DOCX)

**S4 Table. Probability of being non-farm employed for women and men aged 25–55 in rural Malawi, Tanzania and Nigeria.**
(DOCX)

**S5 Table. Probability of being non-farm employed for women and men aged 25–55 in urban Malawi, Tanzania and Nigeria.**
(DOCX)

**S6 Table. Earnings functions of non-farm employment for men, women and pooled sample in rural Malawi, Tanzania and Nigeria with log hourly pay (real int. $) as dependent variable, corrected for selection bias using Heckman selection models.**
(DOCX)

**S7 Table. Earnings functions of non-farm employment for men, women and pooled sample in urban Malawi, Tanzania and Nigeria with log hourly pay (real int. $) as dependent variable, corrected for selection bias using Heckman selection models.**
(DOCX)

**S8 Table. Kitagawa-Oaxaca-Blinder decomposition of gender pay gap for non-farm employed people aged 25–55 in Malawi, Tanzania and Nigeria, corrected for selection bias using Heckman selection models.**
(DOCX)

**S9 Table. Kitagawa-Oaxaca-Blinder decomposition of gender pay gap at the median for non-farm employed people aged 25–55 in Malawi, Tanzania and Nigeria.**
(DOCX)

**S1 Dataset. Nationally representative data for men and women aged 25–55 in rural and urban Malawi, Tanzania and Nigeria.** Data are derived from LSMS-ISA surveys: Malawi Integrated Household Survey 2016–2017 and Integrated Household Panel Survey 2016; Tanzania National Panel Survey 2012–2013; and Nigeria General Household Survey Panel 2015–2016. These data are publicly available on http://surveys.worldbank.org/lsms/programs/integrated-surveys-agriculture-ISA.
(DTA)

**S1 File. Descriptive statistics.** Stata do-file to construct Tables 1–3.
(DO)

**S2 File. Decomposition.** Stata do-file to construct Table 4, S2 and S3 Tables.
(DO)

**S3 File. Heckman selection.** Stata do-file to construct S4–S8 Tables.
(DO)

**S4 File. Median pay gap.** Stata do-file to construct S9 Table.
(DO)

**S5 File. Robustness checks.** Stata do-file for additional robustness checks.
(DO)

## Acknowledgments

We thank seminar participants at Wageningen University & Research, UCLouvain and KULeuven for their feedback, and are grateful for the comments provided by two anonymous reviewers.

## Author Contributions

**Conceptualization:** Goedele Van den Broeck, Talip Kilic, Janneke Pieters.

**Formal analysis:** Goedele Van den Broeck.

**Writing – original draft:** Goedele Van den Broeck.

**Writing – review & editing:** Talip Kilic, Janneke Pieters.

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
