## [Decision Letter · Decision Letter 0]

17 Feb 2022

PONE-D-21-36418Structural transformation and the gender pay gap in Sub-Saharan AfricaPLOS ONE

Dear Dr. Van den Broeck,

Thank you for submitting your manuscript to PLOS ONE. After careful consideration, we feel that it has merit but does not fully meet PLOS ONE’s publication criteria as it currently stands. Therefore, we invite you to submit a revised version of the manuscript that addresses the points raised during the review process. We have some concerns about the organization, the motivation and the methods that should be addressed before the resubmission.

We look forward to receiving your revised manuscript.

Kind regards,

José Alberto Molina

Academic Editor

PLOS ONE

Journal Requirements:

a) Did participants provide their written or verbal informed consent to participate in this study?

“This research was financially supported by the Research Foundation Flanders (FWO) (FWO post-doctoral fellowship).”

“G.V.d.B. was financially supported by the Research Foundation Flanders (FWO) (FWO post-doctoral fellowship). The funders had no role in study design, data collection and analysis, decision to publish, or preparation of the manuscript.”

Reviewers' comments:

Reviewer's Responses to Questions

**Comments to the Author**

1. Is the manuscript technically sound, and do the data support the conclusions?

Reviewer #1: Yes

Reviewer #2: Partly

2. Has the statistical analysis been performed appropriately and rigorously? 

Reviewer #1: Yes

Reviewer #2: Yes

3. Have the authors made all data underlying the findings in their manuscript fully available?

Reviewer #1: Yes

Reviewer #2: No

4. Is the manuscript presented in an intelligible fashion and written in standard English?

Reviewer #1: Yes

Reviewer #2: Yes

5. Review Comments to the Author

Reviewer #1: Structural transformation and the gender pay gap in Sub-Saharan Africa.

The study is well motivated, the approach followed is appropriate and the data base used is very informative. The paper and the results obtained are interesting for the potential reader.

Having said this, I think the article would benefit from a re-ordering of the sections and tables since structure is a bit messy.

I do not understand why Tables 1 and 2 are in the main text and the rest of tables are in Supplementary Annexes. What is the relevance of Table 1 to be in the main text? It seems to me that the most important table is S10 (in fact, the two figures are derived from that table, which captures the results of the Oaxaca-Blinder decomposition). Tables S1 and S3 are auxiliary; Table S2 seems to be useless (unless authors provide some convincing arguments); Tables S4 and S5 on selection into non-farm employment as robustness checks are presented even before any estimation is carried out; descriptive tables S6 and S7 are presented after regressions; and so on….

In my opinion, a more sensible organization of the paper would be something similar to this:

1. Introduction---

2. Materials and methods (Data, Measurement of hourly pay, and perhaps decomposition analysis---

3. Characteristics of NF-E, and Data description with Tables S6 and S7 (maybe Table 1 if authors consider it)

4. The decomposition analysis with Tables S8 and S9 (may be in Annexes) to present the wage regressions, and then Table 2 and Table S10.

[Aside] Authors should notice that data in Table 2 are sample data and in S10 are estimated. It makes a bit difficult to interpret the mean pays in Table S10 as when compared with table 2. If Average rural pay is 2.61 for men and 1.67 for women, then the difference is 0.94; could not be easier to split that difference between explained and not explained (something like 0.62 explained and 0.32 unexplained)? It’s a bit messy to compute exp(0,528), obtain 1,70, and then saying that women earn 70% less than men.

5. Now, the Robustness checks with results of Table S13 (tables S4, S5, S11 and S12 can still be in the Annexes).

Other comments

Some questions should be addressed in the study

1. Why the choice of this three countries? Data availability?, different stage in structural transformation?

2. Is different stage in structural transformation granted only by different GDP growth and urbanization rate? One would need more reasons to argue that these countries are in different stages in structural transformation ….one doesn’t know the starting point

3. In the robustness checks the authors control for selection into NF-Employment but no into non-participation. Would it be important to control for this?

4. Regarding the choice of selection variables, it is said that variables like marital status and number of children affect employment but not hourly pay. There is various evidence showing that married men earn more than married women, and having children reduce earnings of women. Authors should rely in other variables as a choice for controlling selection (in fact, the authors find that the lambda term is non-significant).

5. From data in Table 1, one deduces that farm-employment in rural Malawi and Tanzania is about 50%, which are not considered in the study. It may affect to differences in pay in a marked way.

6. Why survey recall period of 12 months is the main specification (line 215) instead of 7 days (which is usual in Europe)?

Minor comments:

References are needed for the assertion in lines 53-56

Why is not explicitly cited the seminar work by Oaxaca (1979) and Blinder (1979) (lines 104-9, 189-97)?

Line 257 it lacks “and only for 50% in urban Malawi”.

Reviewer #2: The paper titled "Structural transformation and the gender pay gap in Sub-Saharan Africa" analyzes the gender pay gap in Malawi, Tanzania and Nigeria in recent years and relate it with the structural stransformation of the labour market and progressive urbanization of the region. The paper is well written, the methodology applied is standard but mostly well executed (see comments below), and the analysis of the gender pay gap in less developed countries and its evolution along their development path is certainly a topic that deserves attention.

To me there is one important question to be clarified, that is the way in which the link between structural transformation and gender pay gap is analyzed. The paper claims that the three countries under investigation are on different level of structural transformation, so analyzing differences between countries can give information about how the structural transformation is related with the evolution of gender pay gap. However, given the many substantial differences between these three countries, claiming that differences in the pay gap may be explained by the different level of structural information is not reliable. The alternative to compare urban vs rural areas also is not actually indicative of the structural trasformation process alone, also because the definition of ruiral vs urban areas is not fully consistent in between countries. So I would be more careful in interpreting the results in terms of structural transformation.

To improve the analysis of the aforementioned link, one possible empirical strategy could be that the gender pay gap in each country was analyzed at different points in time. This is certainly possible for Malawi as similar data is avbailable since the late nineties. This could be done to a lesser extent in Nigeria, although similar data started only in 2006 (if I remember well), unsure for Tanzania. The authors could decide to focus just on one country or two, depending on data availability and relevance of the structural transformation progress.

Other Empirical points.

1) For all three countries more recent data are available. It would be a nice plus to update the results to the latest available data.

2) It seems that the results for the pooled sample are mostly driven by Nigeria, which is not surprising given the population difference. That said, and given the concerns about differences between countries, the paper would be equally interesting without presenting the results for the pooled sample.

3) Hourly earnings may be a quite volatile measure, for instance for self employed it could be specially low because the declared work hours is generally very large. Although it is quite standard for developed countries, I wonder whether for a developing country using weekly earnings and controlling for the working hours for the decomposition could be an option? That could also allow to avoid winsorizing the distribution.

4) For the sample selection model, the choice of the selection variables seems rather unusual. Besides marital status, which clearly influence the work status (besides being in non-agricultural work), the others are all variables that depend on the choice of being in agricultural job or not. For instance, it is more important for agricultural workers to have a large number of children. Of course having landholding and livestock are clearly influence by the choice of being in agricultural work (ex post). Than other exclusion restrictions would be needed to make the sample selection correction more credible.

6. PLOS authors have the option to publish the peer review history of their article (what does this mean?). If published, this will include your full peer review and any attached files.

Reviewer #1: No

Reviewer #2: No

---

## [Decision Letter · Decision Letter 1]

19 Sep 2022

PONE-D-21-36418R1Structural transformation and the gender pay gap in Sub-Saharan AfricaPLOS ONE

Dear Dr. Van den Broeck,

Thank you for submitting your manuscript to PLOS ONE. After careful consideration, we feel that it has merit but does not fully meet PLOS ONE’s publication criteria as it currently stands. Therefore, we invite you to submit a revised version of the manuscript that addresses the points raised during the review process.

I am particularly agree Rev 1. It is necessary to address both indicated aspects, specifically the first one.

We look forward to receiving your revised manuscript.

Kind regards,

José Alberto Molina

Academic Editor

PLOS ONE

Journal Requirements:

Additional Editor Comments:

I am particularly agree Rev 1. It is necessary to address both indicated aspects, specifically the first one.

Reviewers' comments:

Reviewer's Responses to Questions

**Comments to the Author**

1. If the authors have adequately addressed your comments raised in a previous round of review and you feel that this manuscript is now acceptable for publication, you may indicate that here to bypass the “Comments to the Author” section, enter your conflict of interest statement in the “Confidential to Editor” section, and submit your "Accept" recommendation.

Reviewer #1: (No Response)

Reviewer #2: All comments have been addressed

2. Is the manuscript technically sound, and do the data support the conclusions?

Reviewer #1: Yes

Reviewer #2: Yes

3. Has the statistical analysis been performed appropriately and rigorously? 

Reviewer #1: Yes

Reviewer #2: Yes

4. Have the authors made all data underlying the findings in their manuscript fully available?

Reviewer #1: Yes

Reviewer #2: Yes

5. Is the manuscript presented in an intelligible fashion and written in standard English?

Reviewer #1: Yes

Reviewer #2: Yes

6. Review Comments to the Author

Reviewer #1: I think the article has benefited from the reordering, now being much easier to read, understand and follow. Most of the issues I pointed out in the first review have been satisfactorily addressed by the authors.

However, in my opinion there are still two questions whose answers have not completely satisfied me, and I would like to ask for some additional clarification

First, the authors claim that countries are at different stages of structural transformation according to indicators on GDP per capita, non-agricultural employment, and urbanization rates. Indeed, based on these data, it appears that Nigeria is "ahead" of Tanzania and Malawi. However, we do not know the starting point of each of the countries. We do not know how they were in the year 2000, which is the date that the authors consider the beginning of the structural transformation. The authors claim that (page 5, lines 353-354), “All three countries have embarked on the path of structural transformation during the past 20 years, albeit at different paces with growth rates highest in Nigeria and lowest in Malawi.” One can think that the countries were in a similar situation in year 2000 and that Nigeria has grown much faster. However, it is most likely that Nigeria was already the most advanced country in 2000. It is not clear from what is said in the text, and this may have implications for considering that they are at different stages of structural transformation.

In addition, in lines 341-342, it says that “We pay specific attention to education, sector and occupation, as structural transformation is typically associated with changes in these dimensions.” Looking at the tables, it can be seen that there are differences between the levels of education in the countries, although not very pronounced, and that there are hardly any differences between the distribution of employment by occupations and by sectors across countries. Therefore, one could infer that there are not so many differences in the stage of structural transformation.

In this respect, one wonders about the importance of oil exports in the Nigerian GDP per capita, and how this could distort the feeling that Nigeria is in a more advanced stage than the other two countries.

All this preamble is to try to say that I don't quite understand the relationship between the stage of structural transformation and that the gender wage gap should decrease sooner in Nigeria than in other countries. In fact, authors say (lines 617-619) “Thus, the gender pay gap is not always lower in urban areas or in countries that are at a more advanced stage of structural change. This implies that structural transformation does not consistently help bridge the gender pay gap.” This is also the case in European countries, for example. The Nordic countries have a more advanced stage than those of the South and, nevertheless, the gender wage gap is not less. For example, due to the strong occupational segregation in the Nordic countries.

A second, less important point, regards to the choice of selection variables, the authors acknowledge that selected variables may not be “perfectly exogenous” to hourly pay estimates. However, the authors maintain their estimates and try to justify them by arguing that according to Wald tests these variables cannot be rejected in the specification. Additionally, they use recentered influence functions as a robustness of their results.

Could the authors present some tests of exogeneity and validity of the instruments, as an alternative?

Minor comment:

Regarding occupations, would it not make sense to include the classification in terms of the one-digit 10 groups of the ISCO classification and not between self-employed and salaried?

Reviewer #2: (No Response)

7. PLOS authors have the option to publish the peer review history of their article (what does this mean?). If published, this will include your full peer review and any attached files.

Reviewer #1: No

Reviewer #2: No

---

## [Author Response · Author response to Decision Letter 1]

24 Oct 2022

Dear,

We included a cover letter and response letter in our resubmission to highlight the changes we made to our manuscript.

Kind regards,

Goedele Van den Broeck

---

## [Decision Letter · Decision Letter 2]

14 Nov 2022

Structural transformation and the gender pay gap in Sub-Saharan Africa

PONE-D-21-36418R2

Dear Dr. Van den Broeck,

We’re pleased to inform you that your manuscript has been judged scientifically suitable for publication and will be formally accepted for publication once it meets all outstanding technical requirements.

Kind regards,

José Alberto Molina

Academic Editor

PLOS ONE

Additional Editor Comments (optional):

Reviewers' comments:

Reviewer's Responses to Questions

**Comments to the Author**

1. If the authors have adequately addressed your comments raised in a previous round of review and you feel that this manuscript is now acceptable for publication, you may indicate that here to bypass the “Comments to the Author” section, enter your conflict of interest statement in the “Confidential to Editor” section, and submit your "Accept" recommendation.

Reviewer #1: All comments have been addressed

2. Is the manuscript technically sound, and do the data support the conclusions?

Reviewer #1: Yes

3. Has the statistical analysis been performed appropriately and rigorously? 

Reviewer #1: Yes

4. Have the authors made all data underlying the findings in their manuscript fully available?

Reviewer #1: Yes

5. Is the manuscript presented in an intelligible fashion and written in standard English?

Reviewer #1: Yes

6. Review Comments to the Author

Reviewer #1: The authors succesfully addressed all the comments I made to previous versions of the paper. It's now ready for cosnideration

7. PLOS authors have the option to publish the peer review history of their article (what does this mean?). If published, this will include your full peer review and any attached files.

Reviewer #1: No

---

## [Editor Report · Acceptance letter]

17 Nov 2022

PONE-D-21-36418R2 

Structural transformation and the gender pay gap in Sub-Saharan Africa 

Dear Dr. Van den Broeck:

I'm pleased to inform you that your manuscript has been deemed suitable for publication in PLOS ONE. Congratulations! Your manuscript is now with our production department. 

Kind regards, 

on behalf of

Professor José Alberto Molina 

Academic Editor

PLOS ONE